# Genetic Diversity and Population Differentiation of Kashgarian Loach (*Triplophysa yarkandensis*) in Xinjiang Tarim River Basin

**DOI:** 10.3390/biology10080734

**Published:** 2021-08-01

**Authors:** Xiaoyun Zhou, Shaokui Yi, Wenhao Zhao, Qiong Zhou, Jianzhong Shen, Dapeng Li, Bin Huo, Rong Tang

**Affiliations:** 1College of Fisheries, Key Laboratory of Freshwater Animal Breeding, Ministry of Agriculture and Rural Affairs, Huazhong Agricultural University, Wuhan 430070, China; zhouxy@mail.hzau.edu.cn (X.Z.); 2019308110039@webmail.hzau.edu.cn (W.Z.); hainan@mail.hzau.edu.cn (Q.Z.); jzhsh@mail.hzau.edu.cn (J.S.); ldp@mail.hzau.edu.cn (D.L.); huobin@mail.hzau.edu.cn (B.H.); tangrong@mail.hzau.edu.cn (R.T.); 2College of Life Sciences, Huzhou University, Huzhou 313000, China

**Keywords:** Tarim River, gene flow, barrier, Bayesian skyline plotting

## Abstract

**Simple Summary:**

The distribution of Kashgarian loach (*Triplophysa yarkandensis*) is limited to the Tarim River basin, which is the largest inland river in China. However, the population size of *T. yarkandensis* has been diminishing, and it is critically endangered in the Tarim River basin due to the gradual depletion of water resources, together with alien invasion and agricultural cultivation in Tarim River. In this study, we adopted the RAD-seq method to investigate the population genetics of *T. yarkandensis*, and a high degree of genetic variations and significant genetic differentiation was detected among *T. yarkandensis* populations in the Tarim River basin. The obtained data contribute to understanding the genetic status of *T. yarkandensis*, and help to provide the scientific management strategies and direct future monitoring and utilization of the genetic resource in Xinjiang region.

**Abstract:**

The distribution of *Triplophysa yarkandensis* is restricted to Xinjiang’s Tarim River basin. We collected 119 *T. yarkandensis* samples from nine geographic populations in the Tarim River basin and utilized the RAD-seq method for SNP genotyping. In this study, a total of 164.81 Gb bases were generated with the Illumina platform, and 129,873 candidate SNPs were obtained with the Stacks pipeline for population genetic analyses. High levels of genetic diversity were detected among nine populations. The AMOVA results showed that the majority of genetic variations originated from among populations (*F*_ST_ = 0.67), and the pairwise *F*_ST_ values ranged from 0.4579 to 0.8736, indicating high levels of genetic differentiation among these populations. The discriminate analysis of principal components (DAPCs) and neighbor joining (NJ) tree revealed that the nine populations could be separated into two clusters (i.e., south and north populations), and modest genetic differentiation between south and north populations was observed, while the individuals from several populations were not clustered together by geographical location. The evidence of two genetic boundaries between south and north populations (except TTM) was supported by barrier analysis. The Bayesian skyline plotting indicated that *T. yarkandensis* populations in the Tarim River basin had not experienced genetic bottlenecks, and the effective population size remained stable. This study first clarified the genetic diversity and differentiation of *T. yarkandensis* populations in the Tarim River basin, and it provided valuable molecular data for conservation and management of natural populations.

## 1. Introduction

*Triplophysa yarkandensis*, belonging to Nemacheilidae, is a kind of small economic fish. It is mainly distributed in slow-moving and shallow water habitats. Specifically, the distribution of *T. yarkandensis* is limited to the Tarim River basin, which is the largest inland river in China. The Tarim River and its tributaries are typical seasonal rivers in an arid area, and it often does not produce runoff. In recent years, climate change and human activities resulted in the water curtailment and river desiccation of the Tarim River basin [1]. As a result, due to the gradual depletion of water resources, together with alien invasion and agricultural cultivation in the Tarim River, the population size of *T. yarkandensis* has been diminishing and it is critically endangered [2]. Changing this trajectory will require coordinated research and conservation strategies to provide a better understanding of population genetics at large spatial scales.

*T. yarkandensis* plays an important role in the fragile local ecosystem of the Tarim River basin, and previous studies have shown that the origin and evolution of the genus of *Triplophysa* may be associated with uplifts of the Tibetan Plateau [3,4,5]. Previously, many studies have reported the distribution, morphological [6] and reproductive characteristics [7], mitochondrial genome [8,9] and transcriptome [2] of *T. yarkandensis*. However, information on population structure and historical population dynamics of *T. yarkandensis* populations in the Tarim River basin is still lacking. Therefore, evaluating the genetic diversity, population structure and population dynamics of *T. yarkandensis* is quite essential for the further utilization of genetic germplasm resources and wild population conservation.

Over the past years, restriction site-associated DNA sequencing (RAD-seq) has become a powerful and widely-used technique in conservation biology studies, especially for the species without a reference genome [10,11,12]. Unlike microsatellites and mitochondrial genes (e.g., *cytb*, *coi*, d-loop), more than thousands of SNPs at genomic level could substantially improve the capacity to unravel genetic diversity and divergence in species with both strong and weak population structure [13]. For instance, the significant genetic differentiation of *Misgurnus anguillicaudatus* populations in the Yangtze River basin was detected with 2092 high-quality SNPs [14], but another study failed to detect these findings using microsatellite and mitochondrial genes [15]. Recently, the RAD-seq technique, including GBS, 2b-RAD and dd-RAD, has been widely used in genetic studies of aquatic species, such as red swamp crayfish [16], black bass (*Micropterus spp.*) [17], *Rhodeus ocellatus kurumeus* [18] and Nujiang catfish (*Creteuchiloglanis macropterus*) [11]. In this study, we used the RAD-seq method to unravel the population dynamics and population structure of *T. yarkandensis* in the Tarim River basin. We aimed to (1) use SNP genotyping to investigate fine-scale genetic variations of *T. yarkandensis* populations; (2) evaluate the population differentiation of the populations in the Tarim River basin and (3) provide genomic evidence on the historical population dynamics of *T. yarkandensis*. These data are essential to providing scientific management strategies and direct future monitoring of the genetic resource of *T. yarkandensis* in Xinjiang region.

## 2. Materials and Methods

### 2.1. Sampling and DNA Extraction

Fish samples were taken from 14 locations in the Tarim River basin using nylon line fishing nets from May to October 2019. In total, 119 specimens were collected from nine sample sites in the Tarim River basin (Figure 1). The tail fin clips from the specimens were obtained and stored in 95% ethyl alcohol. Total genomic DNA was extracted from 119 samples using the QIAamp^®^ DNA Mini Kit (QIAGEN, Hilden, Germany) following manufacturer’s instructions. The RNA pollution was treated using Ribonuclease A (TaKaRa, Dalian, China). DNA quality was assessed by 1.2% agarose gel electrophoresis, and the DNA concentration was determined by the NanoDrop 2000 spectrophotometer (Thermo Scientific, Wilmington, DE, USA) and Qubit 2.0 (Life Technologies, Gaithersburg, MD, USA).

### 2.2. RAD Library Construction and Sequencing

The predicted electronic enzymatic digestion was performed using the genome sequences of a closely-related species, *T. tibetana* [19], as a reference genome, and a combination of two endonucleases (*Rsa*I and *HinC*II) was selected. Genomic DNA was incubated at 37 °C with *Rsa*I, T4 DNA ligase, ATP, and *Rsa*I adapter. Restriction-ligation reactions were heat-inactivated at 65 °C, and then digested for additional restriction enzyme *HinC*II at 37 °C. The PCR reaction was performed using diluted restriction-ligation samples, dNTP, Taq DNA polymerase and primer containing barcode. Subsequently, the PCR productions were purified and pooled. The pooled sample was incubated at 37 °C with *Rsa*I, T4 DNA ligase, ATP and Solexa adapter. The digested fragment sequences with lengths of 364–464 bp were enriched for library construction following the protocol [20]. The PCR products were purified using a Gel Extraction Kit (QIAGEN, Hilden, German), and then the gel-purified products were sequenced on an Illumina HiSeq 2500 system (Illumina Inc., San Diego, CA, USA) with 125 paired-end.

### 2.3. Data Processing

The raw data generated from the Illumina sequencer were filtered with the fastp program [21]. To confirm the reliability of data, we randomly selected 2000 reads and mapped them to the NCBI nucleotide non-redundant nucleotide sequence (NT) database. The Stacks pipeline [22] was used for RAD-seq data processing, SNP discovery and standard population genetic statistics. Given that the reference genome of *T. yarkandensis* is not available, a de novo orthology search was conducted. Briefly, the retained reads were demultiplexed according to the barcodes using the process_radtags program from the Stacks pipeline. The ustacks program was used to align the individuals into exactly matching stacks with the parameter of *M* = 4. A catalog of RAD loci was built with cstacks and all individuals in the population were matched against the catalog using the sstacks program. The populations program was used to filter loci (r = 0.75, min-maf = 0.05, max_obs_het = 1). To address linkage disequilibrium (LD) between SNPs and to account for potential ascertainment bias, only one SNP per RAD locus was randomly selected with the populations parameter of “–write_single_snp”. The test of Hardy–Weinberg equilibrium (HWE) for each population was performed with populations program. To eliminate the bias of loci out of HWE, a Perl script “filter_hwe_by_pop.pl” implemented in dDocent program [23] was used to filter the loci with the parameters of “–h 0.001, –c 0.25”.

### 2.4. Genetic Diversity and Population Structure

The obtained loci were used to evaluate population genetic diversity, including major allele frequency, observed heterozygosity (*H*_O_) and expected heterozygosity (*H*_E_), and nucleotide diversity. Pairwise F-statistics (*F*_ST_) among the nine populations were determined with Arlequin v3.4.5 [24]. Genetic clustering and admixture analyses were performed with the “LEA” package [25] in R. The best-fit number of clusters (*K*) was assessed by calculating an entropy criterion with cross-validation using the snmf function. Lower cross-entropy values indicate a better model fit. Meanwhile, population admixture was also evaluated with a Bayesian approach using fastStructure [26]. The ChooseK.py script was used to determine the optimal *K* value. Meanwhile, we analyzed the genetic structure using the non-model-based multivariate approach DAPC (discriminant analysis of principal components) implemented in the “adegent” package [27]. The optimal number of clusters was determined as the one for which the Bayesian information criterion (BIC) showed the lowest value.

We used plink v 1.90 [28] to perform principal components analysis (PCA), and the scatter plotting was performed with the first and second components using “ggplot2” package [29] in R. We constructed a neighbor joining (NJ) tree to cluster the populations using the “ape” package [30] in R. In addition, the distribution of genetic variations was determined by analyses of molecular variance (AMOVA) using the Arlequin v3.4.5. The genetic barriers associated with each geographical location and population were investigated using Monmonier’s maximum-difference algorithm in BARRIER version 2.2 [31]. The pairwise genetic distances among nine populations were determined by *F*_ST_/(1−*F*_ST_).

## 3. Results

### 3.1. SNP Identification and Genotyping

A total of 169.92 Gb raw data was generated by sequencing of the RAD libraries, with an average of 1.37 Gb bases per individual. After quality control, 164.81 Gb clean data was remained for the subsequent analysis (Appendix A). All of the randomly selected reads mapped to the sequences of fish species (e.g., *Cyprinus carpio*, *Danio rerio*) in the NCBI NT database, which indicated that the obtained reads were reliable. An average of 10.90 million tags were obtained for each individual. After filtering the stacks with low coverage, 186,838 assembled loci were genotyped with an average coverage per individual of 17.10. According to the filtering criterion that loci presented in >75% individuals in the populations and MAF > 0.05, 129,873 candidate SNP markers were obtained for the subsequent genetic analysis (Table 1). With the filter using the populations program, the mean length of the remaining loci was 284.15 bp. Transversions (Tv) outweighed transition polymorphisms (Ti), accounting for 53.58% of the SNP sites, with an observed Ti:Tv ratio of 0.87.

### 3.2. Genetic Diversity and Population Structure

The genetic diversity and allele polymorphism of nine populations were generally consistent. Observed heterozygosity (*H*_O_) across all populations ranged from 0.2260 to 0.2604 within nine populations. *H*_O_ was lowest in WSX (0.2260) and highest in JDT (0.2604). The nucleotide diversity (*P*i) among nine populations ranged from 0.2364 to 0.2861, which was highest in KZE. The individual inbreeding coefficients (*F*_IS_) of the populations ranged from 0.0236 to 0.0779 (Table 2). The number of loci that varied from 0 to 4056 showed significant deviations from the Hardy–Weinberg equilibrium among nine populations. To investigate the genetic structure of the nine populations, DAPC was performed with inferred genetic clusters (*K* = 1 ~ 9). The DAPC BIC showed the lowest value with *K* = 2 (Figure 2A), and indicated that the optimal clustering solution was two genetic clusters (Figure 2B). Likewise, population admixture analysis was performed with the “snmf” function in the “LEA” package, and the number of clusters (*K*) was set from 1 to 9. The results revealed that the optimal number of clusters was two (*K* = 2) as identified by minimum cross-entrop value. In addition, the loci filtered by HWE test were used for the Bayesian analysis using fastStructure (*K* = 1 ~ 9). The result revealed that the optimal *K* value was 2 by running the ChooseK.py script, which is consistent with the number of clusters determined with “snmf” and DAPC methods. The average admixture proportions of two clusters for each population are shown in Figure 1. Most individuals from TKX, DWQ, KZE and TTM exhibited obviously different ancestry information compared with the individuals in the AKT, SWS, JDT, WSX and WC (Figure 2C). Several individuals from JDT and SWS also had the admixture ancestry component that dominated in TKX, DWQ and KZE.

The NJ tree of the 119 individuals (Figure 3A) features two genetically distinct clusters, with the most individuals from AKT, WC, WSX, and SWS grouping together (cluster 1) and the other individuals clustering into another cluster (cluster 2). The individuals from DWQ together with TKX and KZE individuals were dominant in the cluster 2, which coincided with the results of the structure analysis. Notably, the TTM individuals and some JDT individuals were clustered into cluster 2, and some TTM and JDT individuals were grouped into a subcluster in cluster 2, indicating a relatively high genetic distance between these two populations and other populations in cluster1. Additionally, similar results were observed with the results of PCA plotting (Figure 3B). The PCA showed no significant clustering by populations or geographic locations, but the individuals from TKX, DWQ and KZE were grouped together, which have a smaller geographic distance than other populations. The individuals from AKT, WC and WSX together with some individuals from TTM, JDT and SWS were clustered together, and other individuals from TTM, JDT and SWS showed closer affinity to TKX, DWQ and KZE.

### 3.3. Gene Flow and Genetic Differentiation

Pairwise *F*_ST_ values across nine populations ranged from 0.4579 to 0.8736 (Table 3), indicating that the genetic differentiation among these populations was at very high levels. To trace the genetic variations among the populations, AMOVA was performed (Appendix A). The result showed that the majority of genetic variation was due to between-population differentiation (67.01%) with evidence of significant genetic differentiation (*p* < 0.05). The association between genetic and geographic distance was tested with Barrier software, with improved Monmonier’s method. The result showed that the genetic boundaries between the populations supported the two boundaries (Figure 4). The first boundary was between population TTM and south populations (WC, JDT, SWS, AKT and WSX), and the second boundary was found between north populations (DQW, TKX and KZE) and those south populations.

### 3.4. Population Demographic History

Demographic history of *T. yarkandensis* in the Tarim river basin was reconstructed based on the Bayesian skyline plot using the skyline function in the “ape” package (Figure 5). The optimal estimated epsilon was 8.4882, and the estimate of effective population size through time showed that *T. yarkandensis* did not experience historical population bottlenecks, and that the historical population size of *T. yarkandensis* in the Tarim river basin experienced an expansion between 100 and80 thousand years ago (with estimation of substitution rate = 0.0025). Then, the population size underwent a steady period. After that, the population size experienced a modest increase 40 thousand years ago.

## 4. Discussion

The Tarim River basin is the main distribution area of *T. yarkandensis* natural populations. Comprehensive population genetic studies of the *T. yarkandensis* populations in the Tarim River basin contribute to revealing the genetic diversity of natural populations and evaluating the status of germplasm resource. Actually, most areas of the Tarim River basin belong to a depopulated zone with high altitude. Meanwhile, due to the river diversions and cutoffs of the Tarim River basin, the sample collection was quite difficult for biological researchers. In this study, we spent several months collecting the wild samples of *T. yarkandensis* for the population genetic study in 2019. Unfortunately, no *T. yarkandensis* individual was caught in some sampling locations; thus, several populations in the upper reaches of the Tarim River basin were absent in the present study. More populations from other locations (e.g., Bosten lake, Hotan River and Toxkan River) will be collected for systematic evaluation of genetic resource in our future studies.

Given the lack of whole-genome sequences of *T. yarkandensis*, we selected de novo mapping pipeline with Stacks for SNP identification. For genetic analyses, we used the populations parameter of “–write_single_snp” to trim the outlier SNPs, which could impact the accurateness of genetic structure and genetic differentiation inference. A total of 125,893 high-quality SNPs were detected and used for population genetic analyses in this study. To further eliminate the genetic bias, we filtered the obtained loci using the HWE test, and used the filtered loci to carry out the structure analyses and PCA. Overall, it could be concluded that the nine populations of *T. yarkandensis* exhibited high genetic diversity, and atypical genetic differentiation was observed among these populations. Particularly, the genetic variations of these populations mainly derived from the among-population variations. We found that the pairwise *F*_ST_ values among these populations were very high, indicating that the genetic exchange among these populations of *T. yarkandensis* was very limited compared with other small bottom-dwelling fish species, such as *M. anguillicaudatus* [14]. It could be possibly explained by the weak migration abilities in the wild and the fragments of habitats in the Tarim River basin.

According to the geographic locations, we assigned AKT, SWS, WC, JDT, WSX and TTM into south populations, which are distributed in the lower reaches of the Tarim River basin, and DWQ, TKX and KZE were assigned into north populations, which are located in the upper reaches of the Tarim River basin. Based on the results of population structure and the NJ tree, the nine populations were separated into two clusters, but the level of genetic differentiation between two clusters was moderate. Most individuals from south populations were clustered together, and the two clusters could not be completely separated based on the results of genetic admixture and phylogenetic analysis. Notably, most individuals from TTM showed a closer affinity to the north populations. TTM is located in Taitema Lake, which is the terminal lake of the main stream of the Tarim River and the beginning of the Qarqan River. Taitema Lake was dried up and had experienced progressive degradation without water flows over several decades until the ecological water conveyance project (EWCP) was initiated by transferring water from Bosten Lake to the terminal Taitema Lake [32]. We speculated that the implementation of EWCP also accelerated the gene flow from populations located in upper reaches of the Tarim River basin to the TTM population. Actually, aquaculture activities in the Tarim river basin are less developed due to the special environment (high salinity and sediment concentration) [33,34], and the natural populations of *T. yarkandensis* were actually less interfered by agricultural activities. In this context, although anthropogenic factors could affect the genetic structure and differentiation of fish populations [35,36], especially in some economic fish species widely cultured in aquaculture [14,37], we noted that the genetic exchange among most populations mediated by anthropogenic factors was limited in the Tarim River basin. Here, we suggest that the *T. yarkandensis* populations should be monitored in the long term to avoid significant reductions in abundance due to the genetic characterizations, including the high levels of genetic variation originating from among-population variation and the small effective population size. Most importantly, these genetic characterizations together with the fragmented and isolated habits in the Tarim River basin increase the probability of outbreeding depression. The fine-scale genetic differences among populations should be regarded with caution in the conservation of *T. yarkandensis*. Multi-generational outbreeding depression can reduce fitness to a greater extent than reductions generated by inbreeding depression [38]. In this case, although high levels of genetic variation within populations were detected, we suggest joint evaluation of inbreeding and outbreeding risks in the wild in the future conservation and management of this species.

In the past decades, traditional techniques and strategies have been established to monitor biological diversity of fish populations in the wild. With the advance of genome-wide molecular markers, study of fish population genetics is increasingly important and popular in ecological, evolutionary and conservation fields. As mentioned above, *T. yarkandensis* is native to the Tarim River basin, and the wild capture of *T. yarkandensis* rarely occurred. However, biological invasion of other alien fish species in the Tarim River basin, such as *Paramisgurnus dabryanus*, *Perca fluviatilis*, and *Ophiocephalus argus*, severally impact the native fish biodiversity [39]. Actually, the rivers in the lower reaches of the Tarim River basin underwent cutoff in the past decades due to climate change and dam construction in the upper and middle reaches. From May 2000 to November 2001, local government initiated the implementation of EWCP by transferring water from Bosten Lake to the lower reaches of the Tarim River [40], which brings many positive direct effects, such as elevated groundwater level, vegetation coverage recovery and agricultural irrigation; however, this project still have some adverse effects, especially including alien aquatic species invasion. Those alien fish species can be well adapted to the habitats of the Tarim River basin and further result in the decline of native fish species by predation and competition. According to the results of a fisheries resource survey in 2019 (unpublished work), we found that the composition of fish communities in the lower reaches was numerically dominated by small fish species, such as *Pseudorasbora parva*, *Carassius auratus*, and *Hypomesus olidus*, which was different to the upper reaches. Therefore, scientific control of alien fish should be implemented to protect the native fish species in the Tarim River basin, especially in the lower reaches. Meanwhile, artificial breeding and releasing could be adopted as conservation and management approaches for *T. yarkandensis*. Importantly, with the releasing of artificially produced offspring into wild, the genetic consequences of these releases on natural populations should be evaluated to maintain genetic diversity and avoid degeneration of genetic resources in further studies. Overall, comprehensively investigating the genetic diversity and population structure contributed to understanding the genetic status of natural resources of *T. yarkandensis* in the Tarim River basin, and this study laid a foundation for the conservation and management of natural *T. yarkandensis* populations.

## 5. Conclusions

In this study, a total of 129,873 genome-wide SNPs of *T. yarkandensis* were obtained with the Stacks pipeline for population genetic analyses. High levels of genetic diversity were detected among nine populations in the Tarim River basin. The majority of genetic variations originated from among-population variations, and a high levels of genetic differentiation among these populations was observed. The nine populations could be separated into two clusters (i.e., south and north populations), and modest genetic differentiation between south and north populations was observed, while the individuals from several populations were not clustered together by geographical location. *T. yarkandensis* populations in the Tarim River basin had not experienced genetic bottlenecks, and the effective population size remained stable. This study provides useful genetic information for conservation and management of *T. yarkandensis* populations in the Tarim River basin.

## Figures and Tables

**Figure 1 biology-10-00734-f001:**
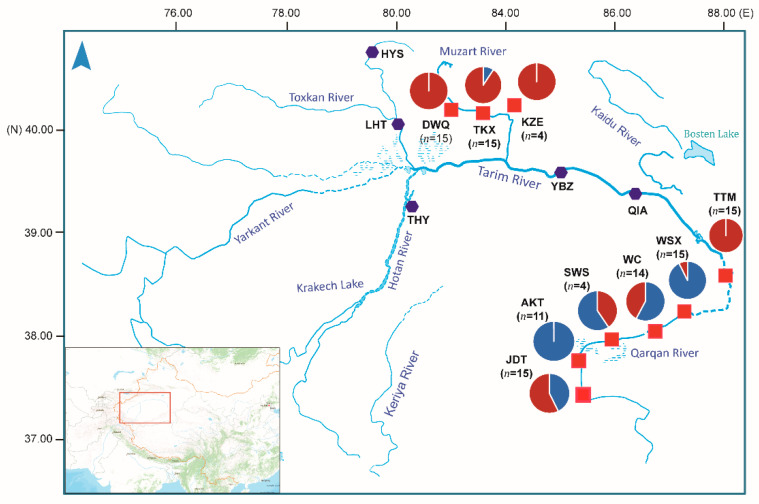
Sample sites of T. yarkandensis in the Tarim River basin located in Xinjiang. The hexagon symbol colored in dark purple represents the sampling locations where no specimen was collected. The red square symbols indicate the sampling locations used in this study. The pie in the map represents average admixture proportion of two genetically distinct clusters estimated with fastStructure.

**Figure 2 biology-10-00734-f002:**
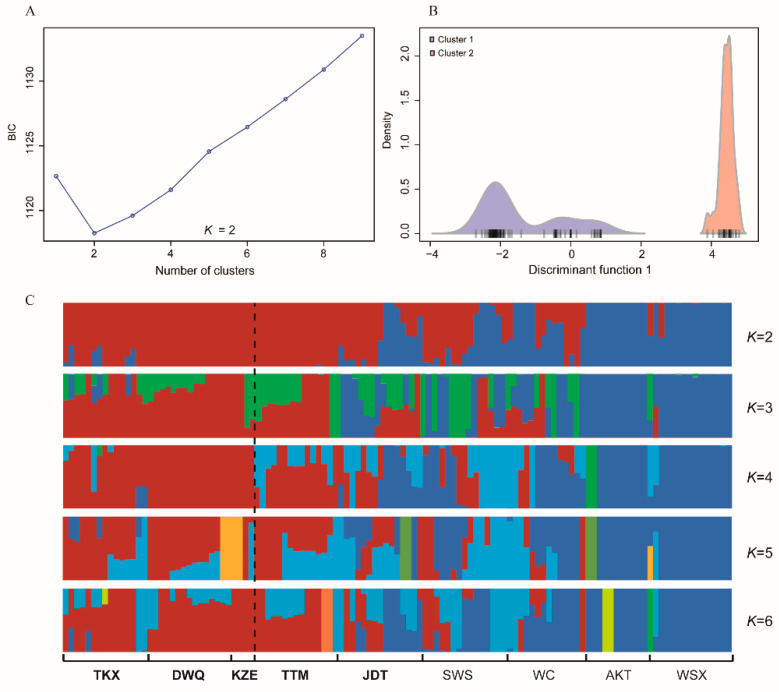
The distribution of BIC values with number of clusters ranged from 1 to 9 (**A**) and the densities of individuals on a given discriminant function for two clusters (**B**); the admixture clustering results for K = 2~6 with Bayesian analysis using fastStructure based on the filtered dataset (**C**). The dashed line represents the separation of south and north populations.

**Figure 3 biology-10-00734-f003:**
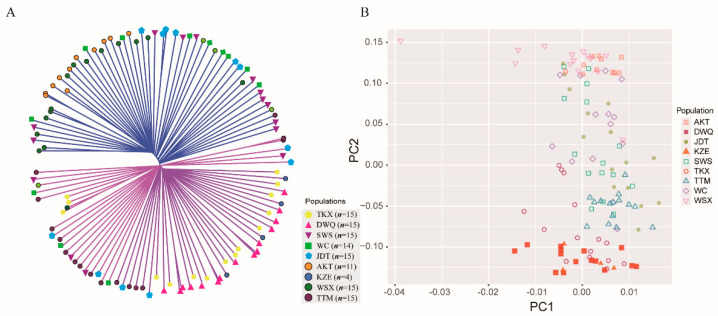
The NJ tree of 119 individuals constructed with the ape package (**A**) and the PCA plotting of the individuals with first and second principal components (**B**).

**Figure 4 biology-10-00734-f004:**
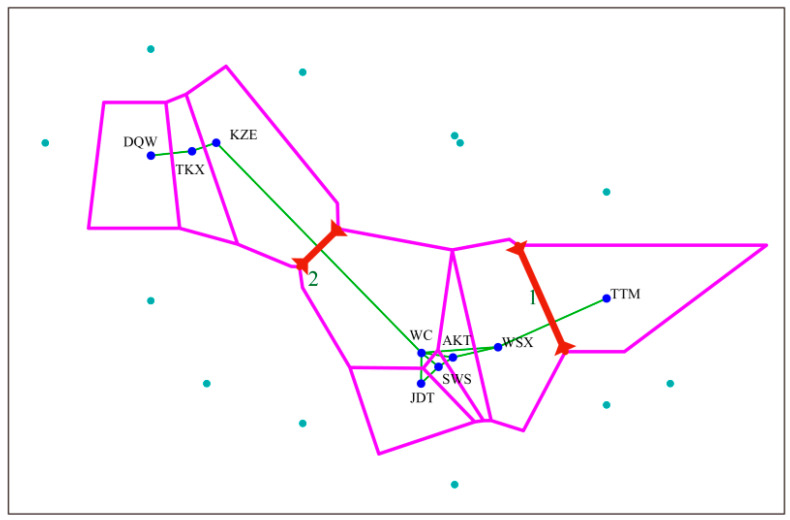
The two genetic boundaries (red bold lines) detected by BARRIER version 2.2 using genetic distance (*N*m). Pink lines and solid green lines represent the Voronoi tessellation and the Delaunay triangulation, respectively.

**Figure 5 biology-10-00734-f005:**
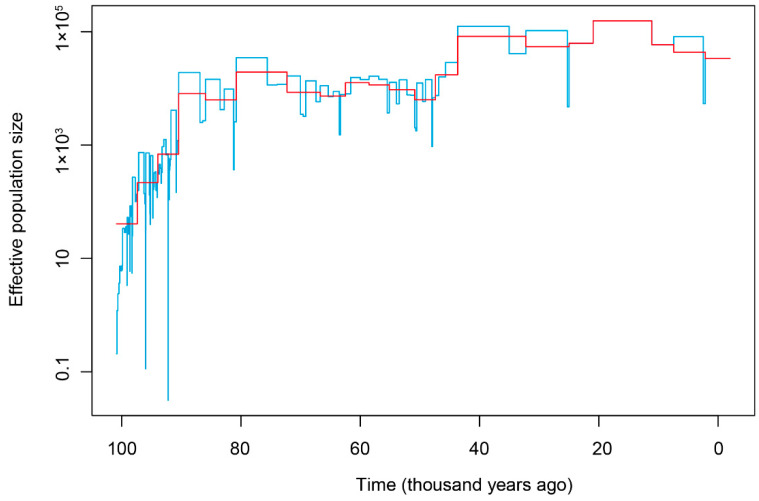
The Bayesian skyline plot of *T. yarkandensis*. The estimate of time unit was based on the substitution rate = 0.0025.

**Table 1 biology-10-00734-t001:** The summary of loci identified with the Stacks pipeline among nine populations.

Population ID	Num of Samples	Samples Per Locus	All Sites	Variant Sites	Polymorphic Sites	Fixed Alleles
TKX	15	12.091	6,825,098	12,019	10,346	32
DWQ	15	13.407	25,761,023	58,354	50,117	1040
SWS	15	13.570	31,271,934	69,585	5955	739
WC	14	12.552	31,093,652	71,003	66,518	1033
JDT	15	13.624	33,647,835	77,904	73,819	1406
AKT	11	10.054	33,397,250	74,608	63,564	839
KZE	4	3.552	40,984,502	97,754	66,521	502
WSX	15	12.424	5,779,906	10,335	8984	9
TTM	15	13.532	25,175,340	57,971	53,848	673
Total	119		54,958,861	129,873		

Note: Fixed alleles indicate the private alleles only detected in a population.

**Table 2 biology-10-00734-t002:** The genetic diversity and allele polymorphism of nine populations.

Pop ID	*p*	Obs_Het	Obs_Hom	Exp_Het	Exp_Hom	*Pi*	*F*is	HWE
TKX	0.8372	0.2356	0.7644	0.2345	0.7655	0.2446	0.0292	372
DWQ	0.8227	0.2462	0.7538	0.2508	0.7492	0.2607	0.0441	3299
SWS	0.8118	0.2562	0.7438	0.2686	0.7315	0.2789	0.0645	3606
WC	0.8107	0.2526	0.7474	0.2689	0.7311	0.2802	0.0779	3766
JDT	0.8078	0.2604	0.7396	0.2725	0.7275	0.2830	0.0667	4056
AKT	0.8244	0.2554	0.7446	0.2492	0.7508	0.2625	0.0236	1989
KZE	0.8195	0.2488	0.7512	0.2445	0.7555	0.2861	0.0715	0
WSX	0.8445	0.2260	0.7740	0.2269	0.7731	0.2364	0.0403	447
TTM	0.8139	0.2516	0.7484	0.2649	0.7351	0.2751	0.0714	3235

Note: *p* indicates mean frequency of the most frequent allele at each locus in this population. Obs_Het and Exp_Het represent mean observed and expected heterozygosity in this population, respectively. *Pi* indicates mean value of π in this population. HWE indicates the number of loci found to be significantly out of Hardy–Weinberg equilibrium (*p* < 0.05).

**Table 3 biology-10-00734-t003:** Pairwise *F*_ST_ among nine populations of *T. yarkandensis*.

*F* _ST_	TKX	DWQ	SWS	WC	JDT	AKT	KZE	WSX	TTM
TKX	0								
DWQ	0.7514	0							
SWS	0.7728	0.5961	0.0000						
WC	0.7720	0.6102	0.4823	0.0000					
JDT	0.7831	0.6126	0.4811	0.4579	0.0000				
AKT	0.8110	0.6397	0.5278	0.5318	0.4965	0.0000			
KZE	0.8617	0.6616	0.5944	0.5630	0.5294	0.5987	0.0000		
WSX	0.6625	0.7732	0.7826	0.7799	0.7903	0.8186	0.8736	0.0000	
TTM	0.7656	0.6207	0.5622	0.5957	0.5773	0.5976	0.6604	0.7719	0

## Data Availability

The data presented in this study are available in this manuscript and its Appendix A and are available on request from the corresponding authors.

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
