# Peer review of "Genetic Diversity and Population Differentiation of Kashgarian Loach (Triplophysa yarkandensis) in Xinjiang Tarim River Basin"

_biology, 2021, doi:10.3390/biology10080734_

Round 1

Reviewer 1 Report

This manuscript seeks to investigate genetic structure among populations of an endangered endemic fish, the Kashgarian loach, in the Tarim River Basin. The authors use RADseq to collect data on almost 130,000 SNPs 'scattered' throughout the entire genome of the loach and find that even neighboring populations are genetically very different from each other - high Fst values suggest little mixing. In contrast, although each population is highly differentiated from the others, there is no evidence that any of these populations are inbred - based on the low Fis values. This is great, for both the manuscript/study, and for potential conservation and management of the endemic loach, if indeed that is the ultimate objective here.

However, I have a handful of problems with the manuscript in its current format. First, the authors need to add more detail to the description of their RADseq library prep and the initial Illumina sequence data processing/filtering. Second, the authors should expand on the discussion of their finding in terms of what they mean for conservation. This appears to be the main reason for the study, so it should be addressed in detail - perhaps some recommendations for how genetic variation in this species can be managed for conservation of the species. Perhaps neighboring populations are subject to high levels of outbreeding depression? What does that mean for conserving the species?

Finally, and most importantly, the manuscript would benefit greatly if it were thoroughly revised by a proficient English speaker. Currently, it is not always clear what the authors are trying to say.

Reviewer 2 Report

Dear authors,

The manuscript entitled “Genetic diversity and population differentiation of Kashgarian loach (Triplophysa yarkandensis) in Xinjiang Tarim River basin” is a well designed population genetic study that aims to understand the levels of genetic diversity and population differentiation of an endemic loach species. To achieve this, the authors used a RAD-seq approach to obtain SNP data at genomic level, and sampled nine populations across the Tarim river basin. Although the number of individuals per sampling site is quite low principally in SWS population (n=4) (usually it is recommended between 20-30 individuals), the difficulties to sample are comprehensively explained in the discussion section. Overall, I had a good impression of this manuscript, although there are some points that should be improved. The main issue that arise to me during the lecture was the interpretation that the authors gave to the levels of genetic differentiation and population structure. Looking to Figure 2C and Figure 3B, I have doubts about the abstract statement “ indicating that the gene flow among these populations was at very low levels”. I think that some clarifications are needed here. For me it seems clear some level of admixture, although it is difficult to interpret both Figures. I would like to see the results of Bayesian analysis performed on fastStructure. Also, it would be useful to plot on map the average membership proportion of each cluster by sampling site, allowing a better phylogeographic interpretation. Beside this, I will provide below some minor comments and some clues to improve the manuscript as whole.

Minor comments:

Figure1: I would like to see the sampling sites in which the authors have tried to sample but not succeed, as they explained on the discussion section.  In addition, I suggest to thick the line of Tarim river to better understand its extent.

2.2. RAD Library construction and Sequencing: I have a curiosity: why the authors used Illumina HiSeq 2500 system with 125 paired end, when the PCR enriched fragments had between 364-464bp?

2.3. Data processing: Did the authors check for Hardy-Weinberg equilibrium across loci?  

Table 1: I would like to see at the end of table a line with the total/average.  I would like to see an explanation of term “fixed-alleles”.

Figure 2: It is very hard to read the figure, since the white bars do not matched with the division line below.

Figure 3: Population colors are very similar and do not allow a clear interpretation. Perhaps, using symbols instead colors would help to distinguish populations.  

Discussion: The results of population structure seems do not completely overlap with the spatial distribution of populations, particularly on TTM, WSX, WC, SWS, AKT and JDT group.  Although the authors raised the possibility of anthropogenic movement of individuals, which may explain the levels of observed admixture, why AKT population remains a single entity when it is located within JDT and SWS, two admixture populations?

Round 2

Reviewer 2 Report

Dear authors,

I appreciated your clarifications and improvements. However, there are still some information and analyses missing. I had suggested to add a map with the average membership proportion for the best number of clusters obtained on fastStructure, which was not done. The results displayed for DAPC and PCA are not convergent (at least in the way they are presented it is not readable), and the two cluster highlighted are very difficult to explain based on geographic distribution of populations. I recommend the authors to test de HWE for each population separately, and those SNPs that are not in HWE removed from analysis, at least from fastStructure analysis. HW test is not mention on material and methods section and results provided need to be clarified. As the authors named the clusters, north and south based on the two clusters obtained on DAPC, but not in PCA, and looking to sampling sites, it would expectable that segregation between populations would be mostly explained by geographic distance (one region sample in the north and other in the south), which is not the case. My guess is that something during SNP processing and selection is introducing a bias in some estimations of population differentiation. 

Please explore this in detail in order to get a better picture of genetic differentiation and structure. This aspect is extremely important in term of species conservation.

Best regards,

Round 3

Reviewer 2 Report

Dear authors,

I am very happy to see all the improvements made in your manuscript, particularly the question of genetic structure and population differentiation. Now, it is more clear the genetic pattern with the results of FastStructure plotted on the map, Figure 1. The last concern that still remain for me, is what happened if you conduct the AMOVA, DAPC and NJ with the dataset used on FastStructure (without loci in HW disequilibrium). Genetic diversity estimates might also be affected by this bias! I know that is a lot of work, but perhaps these new analyses can bring new evidence that would help you to better integrate all the results and explain the species history. Also, I suggest to add information about the number of loci that pass the HWE test. 

Best regards,